# Evaluating the Effect of Nano-SiO_2_ on Different Types of Soils: A Multi-Scale Study

**DOI:** 10.3390/ijerph192416805

**Published:** 2022-12-14

**Authors:** Jiayu Gu, Xin Cai, Youqiang Wang, Dahan Guo, Wen Zeng

**Affiliations:** Badong National Observation and Research Station of Geohazards (BNORSG), Three Gorges Research Center for Geo-Hazards of Ministry of Education, China University of Geosciences, Wuhan 430074, China

**Keywords:** nano-SiO_2_, soil reinforcement, clayey soil, sandy soil, curing condition

## Abstract

A rapid growth in the population leads to a large increase in engineering construction. This means there is an inevitability in regard to building on problematic soils. Soil reinforcement becomes an important subject due to the fact that it is a concern for engineers and scientists. With the development of nanotechnology, more and more nanomaterials are being introduced within the practice of soil reinforcement engineering. In this study, the reinforcing effect of novel nanomaterial nano-silica (SiO_2_) applied to different kinds of soils was systematically studied. The nano-SiO_2_-reinforced soil possessed lower final water evaporation loss, and evaporation rates. The nano-SiO_2_ increased the shear strength of clayey soil and sandy soil under both cured and uncured conditions, but the reinforcing effect on clayey soil was more obvious. The addition of nano-SiO_2_ promotes the friction angle and cohesion of clayey soil; further, it also increases the cohesion of sandy soil. The unconfined compressive strength of clayey soil was enhanced by nano-SiO_2_, meanwhile, the nano-SiO_2_-reinforced soil possessed greater brittleness. The microstructure of nano-SiO_2_-reinforced soil is shown via SEM analysis, and the results of X-ray diffraction (XRD) tests show that there are no new mineral components generated during the reinforcing process. It was also found that nano-SiO_2_ possessed little influence on the soil pH value. Adding nano-SiO_2_ will not damage the original chemical environment of the soil. The microstructure of nano-SiO_2_-reinforced soil was observed to prove the results above. In general, nano-SiO_2_ is an excellent soil additive that can improve the mechanical properties of both clayey soil and sandy soil effectively. This research provides more ideas and directions for the purposes of selecting soil reinforcement materials.

## 1. Introduction

The population of the world continues to grow rapidly in the 21st century which, in turn, has led to higher demands in building infrastructure construction. With the expansion of human construction projects, it is inevitable for engineers and architects to encounter certain “problematic soils” which cannot fulfill engineering requirements. The unreinforced “problematic soils” may lead to a series of safety and reliability problems, including ground collapse, ground settlement, and foundation instability. All of these potential problems can threaten the safety of human life, and property [1,2,3]. In order to improve soil geotechnical properties, a series of physical and chemical means were applied in order to improve shear strength, bearing strength, and to reduce settlement and deformation. The exploration of soil reinforcement can be traced back to efforts that were conducted thousands of years ago. The Mesopotamians and the Romans mixed weak soils with limestone or calcium in order to improve the carrying capacity of roads. Further, the Chinese used tree branches to enhance the tensile properties of the soil during the construction of the Great Wall [4]. When cement was invented in 1824, it eventually became (and still is) the most widely used soil improvement material. Further, when cement is used in soil reinforcement, the bearing capacity, shear strength, and unconfined compression strength of soil can be improved substantially [5,6,7]. However, using cement for the purposes of soil reinforcement reduces the workability and durability of soil. In addition, it also adds detrimental effects, such as undegradable toxicity to the environmental conditions [8,9,10]. Moreover, the production of cement contributes to a large amount of greenhouse gas which, in turn, causes global warming. As there is an increasing goal toward stricter carbon neutrality, there is also an urgent requirement to find an environmentally friendly and low-carbon method that can reinforce soil [11,12].

With the development of nanotechnology—due to its great characteristics such as low price, and environmentally friendly production process—nanomaterials are widely used in nearly all natural science fields, including the medicine, environment, energy, and heavy industries [13,14]. Nanomaterials also serve important roles in the area of civil engineering. They are used to reduce the energy consumption of buildings, as well as to improve the stability of soil [15,16]. The nano-magnesia increases the shear strength of seashore soft soil; the UCS (unconfined compressive strength) of clay soil is promoted by nano-silica, and it reaches its peak when at a nano-silica content of 1% [17,18]. Nano-silica can also enhance the shear strength of sandy soil, whereby it improves the cohesion significantly, but has little effect on the internal friction angle. The increase in cohesion can be as much as 343.1% when compared to the unreinforced ones [19]. In cold regions, the mechanical properties of soil are frost-susceptible. Furthermore, the UCS of soil decreases after cycles of freezing and thawing. The nano-silica-stabilized samples show a significant increase in UCS under the same conditions [20]. In addition to being an independent additive for soil consolidation, the nanomaterials can function along with traditional additives. The CBR strength of lime-stabilized soil increases with the addition of nano-ZnO. Moreover, it shows the maximum increase of 25% when at a nano-ZnO content of 1.5% [21]. Adding nano-silica to cement kiln dust reinforces soil, and can lead to soil with considerably more sustainability than cement. The combination of nano-calcium carbonate and carpet waste fibers can also be used to stabilize soil. The soil cohesion can be upgraded up to 1.95 times when compared to the control specimen [22]. Additionally, the use of nano-silica and randomly distributed fibers can promote the UCS of clay soil from 75 kPa to 225 kPa [23]. The nano-particles are characterized by tiny sizes, small degrees of granular distribution, and agglomeration, which make it possible to improve the compactness and structures of soil [24]. In regard to its excellent engineering properties and abundant reserves on Earth, nano-SiO_2_ is being considered to be utilized for improving geotechnical soil properties [25]. According to the research of Patro [26], CBR obtains the maximum value at a nano-silica content of 1%. Additionally, larger nano-silica content may lead to a decrease in soil strength, which is as a result of the uneven distribution of silica nanoparticles, and the agglomeration effect of nanoparticles. As one of the most important parameters in environmental protection, the sustainability of construction material is of great concern. Further, 1% is the optimal value of nano-silica for the highest 28-day UCS. Higher nano-silica addition results in greater costs and higher emissions than a lower 28-day UCS [27]. Therefore, when nano-silica is used for soil reinforcement, the amount of nano-silica addition should be selected carefully in order to achieve the goal of sustainability, in regard to environmental pollution, cost, and soil strength.

The purpose of this study is to evaluate the feasibility of using nano-SiO_2_ for the purposes of soil reinforcement. Previous studies have tended to focus on the reinforcing effect of nano-SiO_2_ on a certain type of soil; thus, the results were not comprehensive. As the representatives of cohesive and non-cohesive soils, clayey soil and sandy soil were selected in order to study the reinforcing effect of nano-SiO_2_ on these two soils. The result can reflect the feasibility of nano-SiO_2_ reinforcement in different soil conditions. At the same time, the mechanical properties of the specimens under different curing conditions were investigated, in order to detail the reinforcing effect of nano-SiO_2_ comprehensively and systematically. A constant temperature evaporation test was conducted in order to study the effect on soil evaporation properties. Direct shear tests and unconfined compressive tests were carried out to test the effect of nano-SiO_2_ on soil mechanical properties. SEM and X-ray diffraction analyses were conducted in order to observe the microstructure of nano-SiO_2_-reinforced soil. The pH value change during the soil curing process was also tested.

## 2. Materials and Methods

### 2.1. Material

Natural soils are insufficient in most engineering applications, as they show properties of low capacity and strength. As the representatives of cohesive and non-cohesive soils, clayey soil and sandy soil are used in this study, respectively. The grain size distribution curves of these two soils are listed in Figure 1. The geotechnical properties of soil are tested according to ASTM D698 (2021) [28], the details of which are listed in Table 1.

The nano-SiO_2_ used in this study, which is a nano-scale silica powder with a white color [29], was produced by Evonic Industries AG. Due to its small particle size, the nano-SiO_2_ has features which include a large specific surface area, strong surface adsorption, high surface energy, and good dispersion performance. Certain basic properties of the nano-SiO_2_ parameters are shown in Table 2. The pictures of clayey soil, sandy soil, nano-SiO_2_ are shown in Figure 2.

### 2.2. Sample Preparation

In order to explore the impact of nano-SiO_2_ content on soil properties, four varying amounts of nano-SiO_2_ content (i.e., 0, 0.5%, 1%, and 2%, calculated via the weight of the dry soil) were, respectively, added to the soil. The maximum additional amount was set as 2%, due to the fact that previous studies have indicated that the optimal addition of nano-SiO_2_ content is often no more than 2% [17,27]. The selected soil was first dried in a 40 °C oven, and then crushed and passed through a 2 mm sieve. It must be noted that the distribution of the small size nanoparticles in the soil may not have been powdered or uniform. In order to make samples homogeneously, the sample preparation procedure follows the steps below [31]. The nano-SiO_2_ was first added to the distilled water (22% for the clayey soil and 11% for the sandy soil, calculated via the weight of the dry soil) and stirred for 30 min by a magnetic stirrer. Then, the homogeneous solution was mixed with the treated soil. After proper mixing, the mixture was preserved in a chamber for 24 h for the purposes of the uniform distribution of nanoparticles and water in the soil. The mixture was placed into cylindrical steel molds with specific dimensions. The samples were statically compacted to 90% of the maximum density of the natural soil. The samples for the constant temperature evaporation test and direct shear test had a diameter of 62.8 mm, and a height of 20 mm. Additionally, the samples for the unconfined compression test had a diameter of 40 mm, and a height of 80 mm. Due to the fact that the curing condition possessed a great influence on the strength properties of the sample [32], the reinforced effect of nano-SiO_2_ on samples under both cured and uncured conditions was tested in this study. Certain prepared specimens were tested after being prepared, which represented the samples under the uncured conditions. Certain other specimens were placed in a ventilated curing room and maintained for 28 days at a temperature of 25 ± 2 °C, which can be regarded as the samples under the cured conditions.

### 2.3. Testing Methods

The constant temperature evaporation test was conducted in order to study the water evaporation qualities of the nano-SiO_2_-reinforced soil. The test was conducted in a specially made thermostatic box (Figure 3) in the laboratory, which can maintain the constant evaporation of the specimen at the preset temperature. The thermostatic box mainly consists of a heating unit, a temperature regulator, and a weight collection system. The specimen was put into the thermostatic box at a constant temperature of 50 °C, and the loss of weight was collected at every minute automatically. Data were recorded until they reached a steady value. Furthermore, the loss of weight was equal to the amount of water evaporation. Direct shear tests and unconfined compressive tests (UCS) were conducted in order to test the mechanical properties of nano-SiO_2_-reinforced soil, under the guidance of ASTM D3080/D3080M-2011 [33] and ASTM D4219-22 [34]. The direct shear tests were conducted at 100, 200, and 400 kPa vertical stress with a shearing rate of 0.8 mm/min, while the unconfined compressive tests are conducted at 2 mm/min. All tests were conducted in triplicate, and the final test results were set as the average of the three valid results. The SEM was conducted in order to investigate the microstructural nature of the nano-SiO_2_-reinforced soil. The samples were cut into sizes of 10 mm × 10 mm × 10 mm and then dried in a 40 °C oven. The dried samples were gold-coated with the arc discharge method, and then tested at accelerating voltages of 20 kV via scanning electron microscope. The X-ray diffraction test was performed in order to explore the change in mineral composition in the nano-SiO_2_-reinforced soil. In regard to the quantitative analysis of the mineral composition, the “K-value” method described in SY/T 5163–2010 [35] was adopted. The Bruker D8 advance was used as the test instrument, and the scanning angle was settled at a range from 5° to 90° with a scanning speed of 10°/min. The soil pH test procedure followed ASTM D4972-19 [36]. Moreover, the test was performed by an AI1201 PH60-E Premium digital pH meter with a glass electrode, and a reference electrode.

To better ensure the true validity of the test results, each group of the above tests was repeated three times. Only when the parallel test results possessed an error under 5% would the data be considered valid, and only then would the final test results be determined as the average of the three valid results.

## 3. Results and Discussion

### 3.1. Constant Temperature Evaporation Test

In arid districts with less rainfall, the rapid evaporation of water may affect the growth of crops. Throughout the history of human development, research on reducing water evaporation loss has been ongoing for this reason [37,38]. Figure 4a shows the cumulative water loss curves of clayey soil. As can be seen, it first increases, and then gradually levels off. Within the same evaporation time, the cumulative water loss of nano-SiO_2_-reinforced clayey soil is smaller than that found in plain soil. Meanwhile, nano-SiO_2_-reinforced clayey soil shows smaller final water evaporation loss than plain soil. The higher proportion, which the nano-SiO_2_ in soil accounts for, the smaller the final water evaporation loss becomes. Figure 4b shows the evaporation rate of nano-SiO_2_-reinforced clayey soil. It can be seen that there are two stages during the evaporation process of soil [39]. The first stage is called the constant-rate stage. In this stage, the evaporation rate remains at a high value. In the second stage, which is called the falling-rate stage, the evaporation rate gradually decreases to zero. In the constant-rate stage, the evaporation rates of nano-SiO_2_-reinforced clayey soil are smaller than the unreinforced one. Additionally, it is inversely proportional to the nano-SiO_2_ content, i.e., the higher the nano-SiO_2_ content that is added, the smaller the evaporation rate becomes.

The cumulative water evaporation loss and evaporation rate curves of sandy soil are shown in Figure 5. From this, it can be seen that they have similar behavior when compared with the clayey soil. The nano-SiO_2_-reinforced sandy soil has a smaller final water evaporation loss, and lower evaporation rates than the plain sandy soil. When compared with clayey soil, sandy soil requires less time to reach its final water evaporation loss. Due to the loose structure and larger porosity of sandy soil, water evaporation in sandy soil is easier and faster than in clayey soil. The initial water content of sandy soil is lower, which causes a smaller evaporation rate. As the evaporation rate is related to porosity and moisture content, the sandy soil has similar evaporation rates to the clayey soil at the constant-rate stage [40].

Due to its small particle size, nano-SiO_2_ has a great specific surface area [30]. Furthermore, the soil reinforced by nano-SiO_2_ will have a larger specific surface area. This means there are more water molecules surrounding the soil particles in the nano-SiO_2_-reinforced soil; thus, the amount of stored pore water is promoted. As a result, the final water evaporation loss and evaporation rates of nano-SiO_2_ -reinforced soil are decreased [41].

### 3.2. Direct Shear Test

The direct shear test results of nano-SiO_2_-reinforced clayey soil are shown in Figure 6. When samples are under uncured conditions (Figure 6a), adding nano-SiO_2_ can significantly improve the shear strength. In addition, the reinforcing effect becomes stronger with the increase in nano-SiO_2_ content. It increases from 70.72, 115.36, and 177.92 kPa to 158.08, 195.56, and 294 kPa, respectively, as the nano-SiO_2_ content increases from 0 to 2% (at the vertical stresses of 100, 200, and 400 kPa). In regard to the samples under cured conditions (Figure 6b), the nano-SiO_2_-reinforced clayey soil still exhibited better strength properties than plain soil. When the nano-SiO_2_ content increased from 0 to 2%, the shear strength of the cured samples increased from 156, 239.2, and 426 kPa to 236.8, 368.4, and 524.6 kPa (at the vertical stresses of 100, 200, and 400 kPa), respectively.

The direct shear test results of nano-SiO_2_-reinforced sandy soil are shown in Figure 7. Moreover, the addition of nano-SiO_2_ has a positive effect on the shear strength of sandy soil. Further, the effect is enhanced with the increase in nano-SiO_2_ content. When the nano-SiO_2_ content increases from 0 to 2, the shear strength of uncured sandy samples increases from 50.3, 107.0, and 206.6 kPa to 64.5, 133.0, and 240.2 kPa, respectively. Additionally, it increases from 70.4, 128.0, and 233.4 kPa to 87.7, 145.6, and 263.9 kPa for cured sandy samples (at the vertical stresses of 100, 200, and 400 kPa), respectively. As such, when compared with clayey soil, the reinforcing effect on sandy soil is more limited.

The shear strength parameters of nano-SiO_2_-reinforced clayey soil were calculated, and are shown in Figure 8. In regard to the samples under the uncured conditions (Figure 8a), with the addition of a small amount of nano-SiO_2_ (0.5%), the friction angle of clayey soil is promoted, while there is little change in soil cohesion. With the increase in SiO_2_ content, the cohesion of clayey soil increases. When samples are under cured conditions (Figure 8b), the soil cohesion increases continuously alongside increases in nano-SiO_2_ content. However, there is little influence on the friction angle for the addition of nano-SiO_2_. As for the shear strength parameters of nano-SiO_2_-reinforced sandy soil (Figure 9), the cohesion of the sandy soil is promoted by the addition of nano-SiO_2_, while the friction angle fluctuates in a small range. However, the addition of nano-SiO_2_ had little effect on the friction angle of the sandy soil.

The particle size of nano-SiO_2_ is much smaller than clayey soil and sandy soil. As such, the nanoparticles can easily disperse in the soil and fill the pores between the soil particles, which, as a result, means that the density of the soil matrix improves, and the effective contact area between soil particles rises [42]. This causes an increase in soil shear strength, and is called the “filler effect” of nano-SiO_2_ [43,44]. At the same time, nano-SiO_2_ can form a colloid in water suspension. When mixed with water in the soil, the nano-SiO_2_ can form a viscous gel to enhance the cementation between soil particles, therefore improving the shear strength [41]. The soil structure becomes compact under the cementing action and flocculation effect of nano-SiO_2_ [45]. As the representative cohesive soil, the cohesion of clayey soil is quite impressive. A small content of nano-SiO_2_ (0.5%) cannot generate enough gel to promote soil cohesion significantly. With the increase in nano-SiO_2_ content, the amount of viscous material formed by nano-SiO_2_ increases. This causes the increase in bonding strength between the soil particles, thus promoting soil cohesion. Due to the loose structure and large particle size of sandy soil, the filling effect is limited at the low addition content level of nano-SiO_2_. Therefore, the reinforcing effect of nano-SiO_2_ on clayey soil is better than on sandy soil

### 3.3. Unconfined Compression Test

The unconfined compression test results are presented in Figure 10. The nano-SiO_2_-reinforced clayey soil possessed a higher unconfined compressive strength than plain soil, in both cured and uncured conditions. With the increase in nano-SiO_2_ content, the unconfined compressive strength of the samples increases significantly. When comparing the compressive strength of the cured and uncured samples, it was found that it increased rapidly from 200–600 kPa to 1500–3000 kPa. In regard to the nano-SiO_2_-reinforced soil, there is a significant loss of strength after reaching the peak strength; the rate of degradation reduces, and finally reaches residual strength [46]. The brittleness index ($I_{B}$) is introduced in order to compare the relationship between the peak strength and residual strength (Formula (1)), which can quantify the ductility of soil [47].
(1)$$I_{B}=1-(\sigma_{r}/\sigma_{\max})$$
$\sigma_{r}$: Residual strength;$\sigma_{\max}$: Unconfined compressive strength.

In regard to the samples under the uncured conditions, the I_B_ increases from 0 to 0.588 with the increase in nano-SiO_2_ content. Meanwhile, the failure pattern of nano-SiO_2_-reinforced clayey soil changed from strain hardening to strain softening [48]. In regard to the samples under the cured conditions, they show greater I_B_ values than the uncured ones, which means an increase in brittleness (Figure 11a). The I_B_ of the cured samples increased as a result of the increase in the nano-SiO_2_ content. This result is similar to that of Kannan’s research [49]. At the same time, as is shown in Figure 11b, the failure strain of nano-SiO_2_-reinforced samples is reduced, as was observed by Kalhor et al. [30]. When nano-SiO_2_ content increased from 0 to 2%, the failure strain of the uncured samples decreased from 10% to 1.28%. Furthermore, it decreased from 2.51% to 1.56% for the cured samples. When mixing with water, the nano-SiO_2_ can form a viscous gel that can enhance the cementation between soil particles. The bonding strength between soil particles is promoted as the nano-SiO_2_ content increases. When samples are damaged by compression, the cementation between soil particles is disrupted, and the soil exhibits brittle failure.

### 3.4. SEM (XRD) and X-ray Diffraction Analysis

The microstructure of nano-SiO_2_-reinforced soil was demonstrated via a scanning electron microscope, the results of which are shown in Figure 12 and Figure 13. The magnification of the clayey soil image is 2000×, while the magnification of the sandy soil image is 400×. When compared to the unreinforced samples, the nano-SiO_2_-reinforced soils have a more compact structure. The nanoparticles fill the pores between soil particles so as to enhance the interfacial contact area and the particle packing density of the soil [19,50]. Meanwhile, the nano-SiO_2_ can form a gel when mixed with water, and during the flocculation and agglomeration process, the cementation between soil particles is enhanced [17]. These two factors influence the frictional properties and cohesion of soil.

The powder diffraction patterns of the plain soil and the 2% nano-SiO_2_-reinforced soil are shown in Figure 14 (these results were gained via the XRD test). There was no obvious difference between the diffraction peak pattern of plain soil, and the nano-SiO_2_-reinforced soil. This meant that during the reinforcing process, no new mineral composition was generated [51]. It must be noted that the main component of nano-SiO_2_ is silicon dioxide, which is similar to natural soil. Having said this, nano-SiO_2_ is generally inert and non-toxic. Thus, nano-SiO_2_ is an environmentally friendly soil reinforcement material.

### 3.5. pH Tests

The pH value of different nano-SiO_2_-content-reinforced soil is presented in Figure 15. Meanwhile, as the most widely used soil reinforcement material, cement-reinforced soil pH was also tested for comparison [52,53]. The pH values of clayey soil and sandy soil were 7.92 and 9.95, respectively. However, they decreased to 7.48 and 8.74 with the increase in nano-SiO_2_ content. In regard to the cement-reinforced clayey soil and sandy soil, the pH values increased significantly to 11.62 and 11.84, which are much higher than for the nano-SiO_2_-reinforced soil. The cement-reinforced soil pH increased profoundly due to the fact that the hydration reaction of cement produces alkaline Ca(OH)_2_ in soil, which directly increases the soil pH value. When compared to the traditional soil stabilization material cement, the nano-SiO_2_-reinforced soil shows smaller pH changes in both sandy and clayey soils. As such, utilizing nano-SiO_2_ for the purpose of soil reinforcement will not damage the original chemical property of the soil.

### 3.6. Limitation and Future Work

There are still some deficiencies in this research which need to be further studied. The soils used are clayey soil and sandy soil, which can be used as typical representatives of cohesive soils, and non-cohesive soils. However, the different mineral composition and formation processes result in the various physical and chemical properties between different soils; thus, the results of this paper may not be suitable for all soils. Further work is still necessary to investigate the reinforcing effect of nano-SiO_2_ on different kinds of soils, especially for special soils with poor engineering properties, for example, soft soil and expansive soil. Future research is needed to explore the engineering applications of using nano-SiO_2_ for soil consolidation, rather than just staying in the laboratory investigation stage. Through the study and demonstration of the applicable conditions, construction process, and economic benefits of nano-SiO_2_ reinforced soil, we provide a theoretical basis and technical support regarding the engineering applications of nano-SiO_2_.

## 4. Conclusions

Due to its excellent properties in regard to soil reinforcement, nano-SiO_2_ is considered to be a potential alternative material to cement. As a result, it is being used in more and more soil reinforcement projects. The proper amount of nano-SiO_2_ can improve soil properties significantly, but the excessive addition of nano-SiO_2_ can also adversely affect the soil strength. When nano-SiO_2_ is used in soil reinforcement, the amount added needs to be carefully considered. In this paper, the improvement effect of nano-SiO_2_ on the physical and mechanical properties of clayey soil and sandy soil was studied. The following conclusions can be obtained:Nano-SiO_2_ has a great specific surface area, which results in a smaller final water evaporation loss and lower evaporation rate in regard to nano-SiO_2_-reinforced soil, when compared with plain soil.Nano-SiO_2_ can improve the shear strength of clayey and sandy soil under cured and uncured conditions. The enhancing effect increases with the increase in nano-SiO_2_ content, and the enhancing effect on clayey soil is better than on sandy soil.Nano-SiO_2_ can fill the pores between soil particles in order to improve the mechanical properties of the matrix. In addition, it can form a gel to enhance the connection between soil particles. By these means, the shear strength of soil is promoted. As sandy soil has larger particles and a looser structure, the reinforcing effect of nano-SiO_2_ possesses less influence.The unconfined compressive strength of nano-SiO_2_-reinforced clayey soil increases with the rise of nano-SiO_2_ content. Nano-SiO_2_ also increases the soil brittleness, which results in a decrease in failure strain.The results of SEM and X-ray diffraction show that nano-SiO_2_ enhances soil strength in two ways: one is by improving the interfacial contact area and the particle packing density of soil; the other is by enhancing the cementation between soil particles. There is no new mineral composition generated in nano-SiO_2_-reinforced soil.When compared with traditional cement material, nano-SiO_2_ has less influence on the soil pH value.

## Figures and Tables

**Figure 1 ijerph-19-16805-f001:**
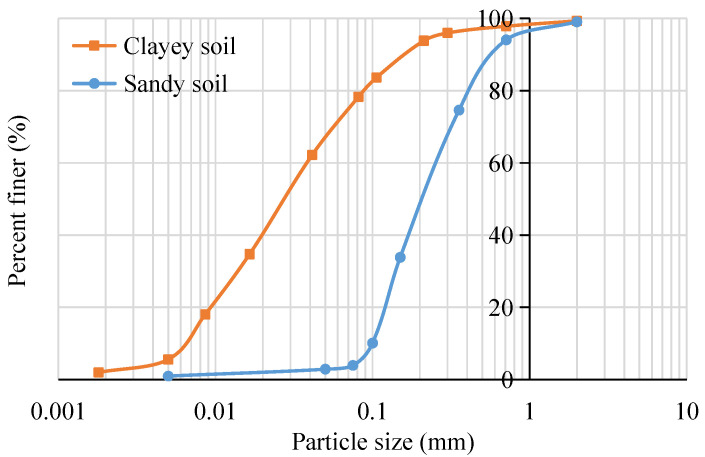
Particle size distribution.

**Figure 2 ijerph-19-16805-f002:**
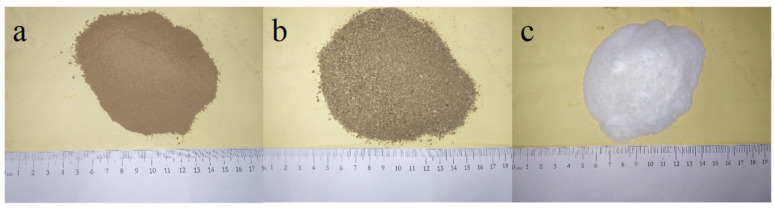
The (**a**) clayey soil, (**b**) sandy soil, and (**c**) nano-SiO_2_ used in this study.

**Figure 3 ijerph-19-16805-f003:**
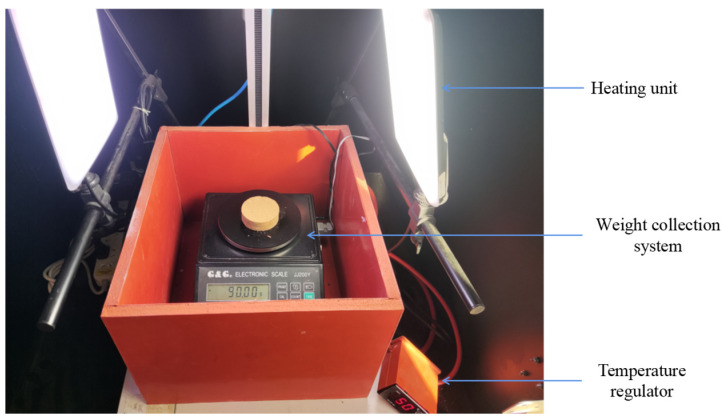
Thermostatic box.

**Figure 4 ijerph-19-16805-f004:**
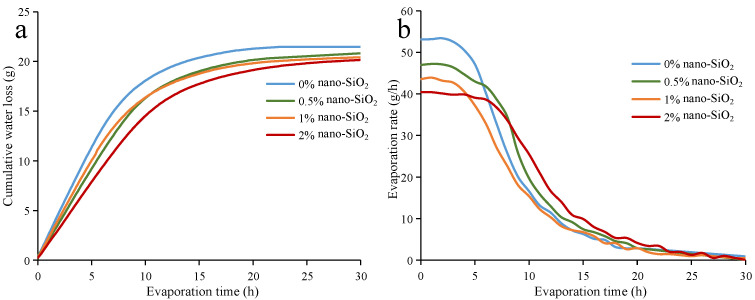
The (**a**) cumulative water evaporation loss and (**b**) evaporation rate of nano-SiO_2_-reinforced clayey soil.

**Figure 5 ijerph-19-16805-f005:**
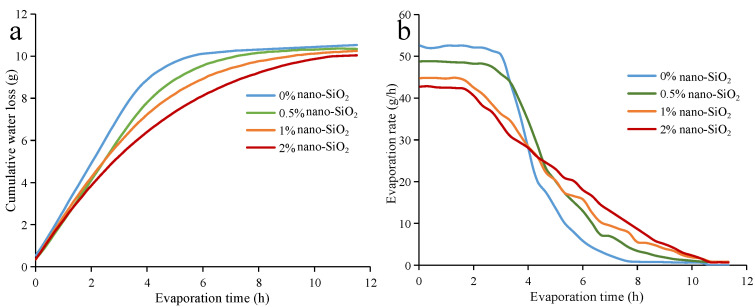
The (**a**) cumulative water evaporation loss and (**b**) evaporation rate of nano-SiO_2_-reinforced sandy soil.

**Figure 6 ijerph-19-16805-f006:**
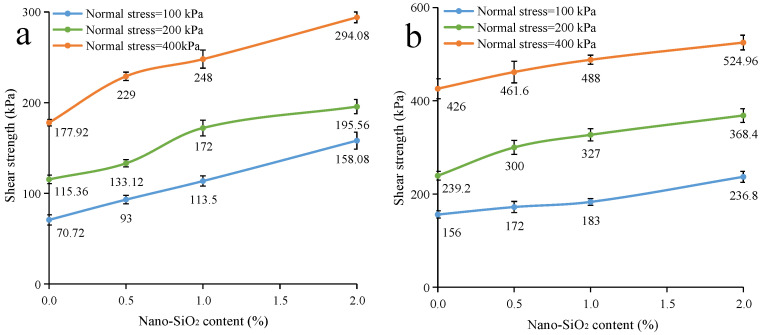
The shear strength of nano-SiO_2_-reinforced clayey soil (**a**) under uncured conditions (**b**) under the cured conditions (Values are means ± S.E.M.).

**Figure 7 ijerph-19-16805-f007:**
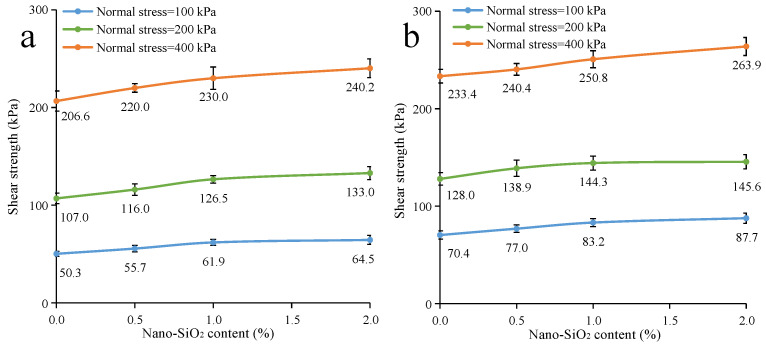
The shear strength of nano-SiO_2_-reinforced sandy soil (**a**) under uncured conditions (**b**) under cured conditions (Values are means ± S.E.M.).

**Figure 8 ijerph-19-16805-f008:**
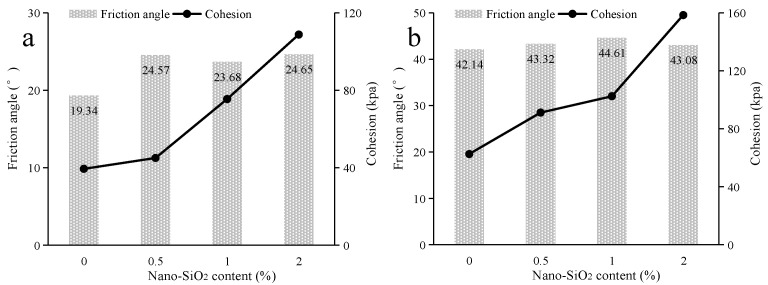
The shear strength parameters of nano-SiO_2_-reinforced clayey soil (**a**) under uncured conditions (**b**) under the cured conditions.

**Figure 9 ijerph-19-16805-f009:**
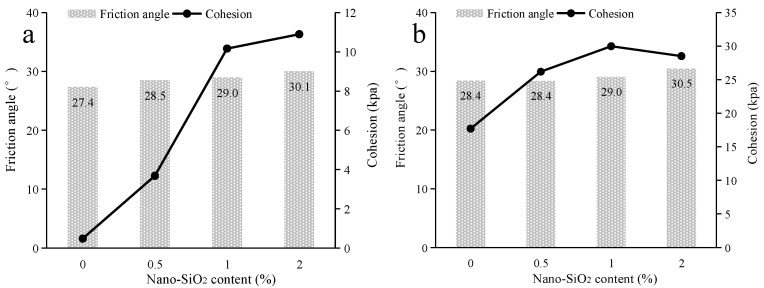
The shear strength parameters of nano-SiO_2_-reinforced sandy soil (**a**) under uncured conditions (**b**) under cured conditions.

**Figure 10 ijerph-19-16805-f010:**
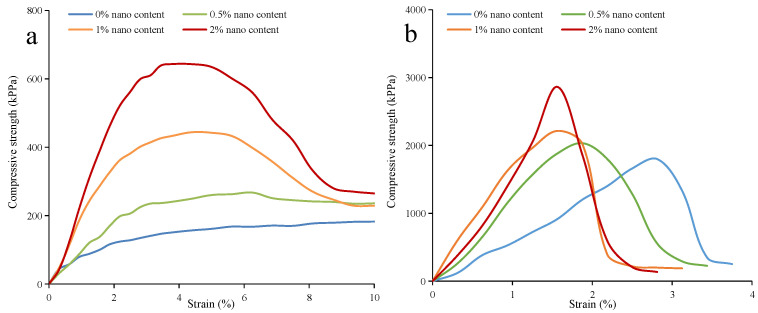
The unconfined compressive strength of nano-SiO_2_-reinforced clayey soil (**a**) under uncured conditions (**b**) under cured conditions.

**Figure 11 ijerph-19-16805-f011:**
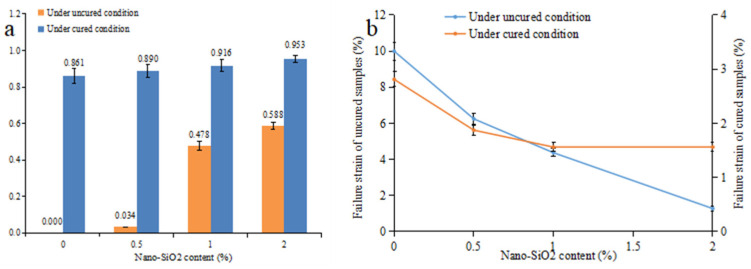
The (**a**) brittleness index and (**b**) displacement in peak shearing strength of the nano-SiO_2_-reinforced clayey soil (Values are means ± S.E.M.).

**Figure 12 ijerph-19-16805-f012:**
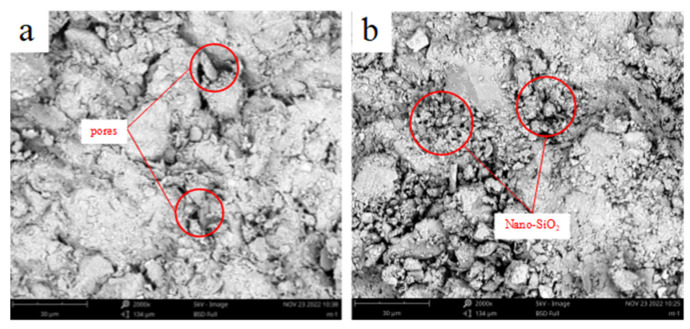
SEM micrographs of the (**a**) clayey soil; (**b**) 2% nano-SiO_2_-reinforced clayey soil.

**Figure 13 ijerph-19-16805-f013:**
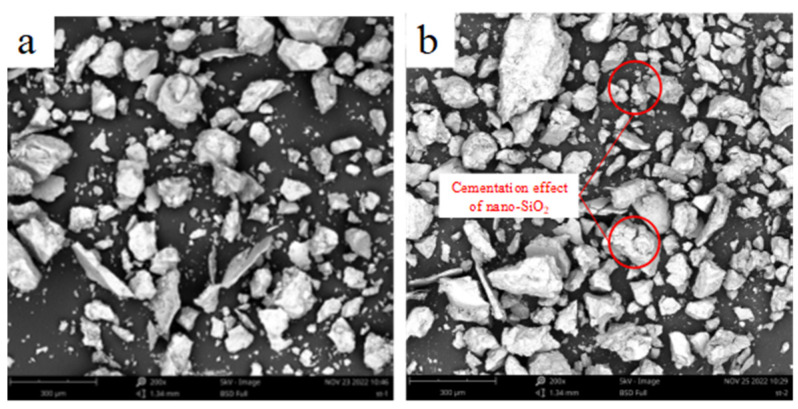
SEM micrographs of the (**a**) sandy soil; (**b**) 2% nano-SiO_2_-reinforced sandy soil.

**Figure 14 ijerph-19-16805-f014:**
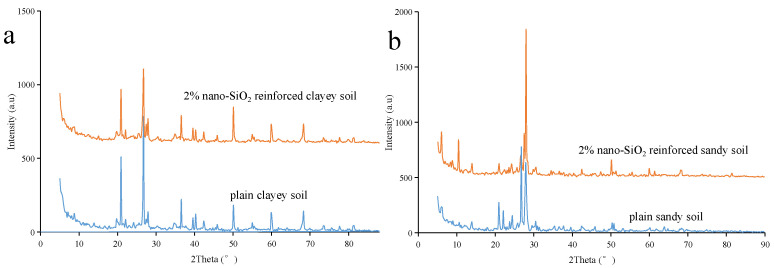
XRD diffraction patterns of the nano-SiO_2_-reinforced (**a**) clayey soil and (**b**) sandy soil.

**Figure 15 ijerph-19-16805-f015:**
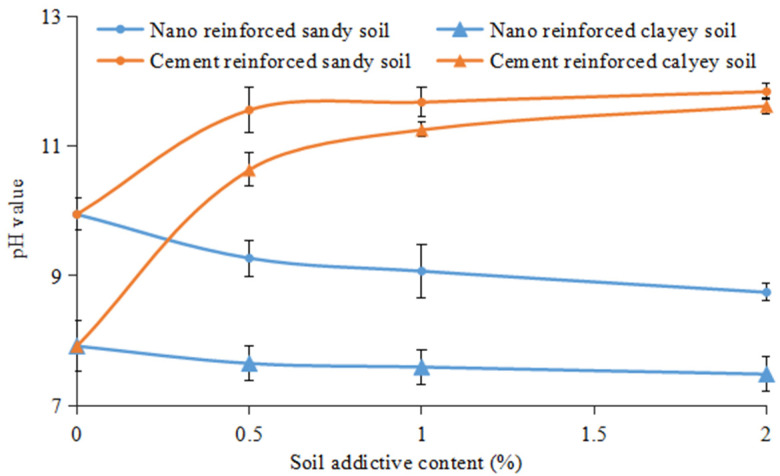
Effect of different additives on the pH value of clayey soil and sandy soil (values are means ± S.E.M.).

**Table 1 ijerph-19-16805-t001:** The basic property indices of soil.

Property	Value	
	Sandy Soil	Clayey Soil
Specific gravity	2.55	2.73
Maximum dry unit weight (kN/m^3^)	17.05	16.85
Optimum moisture content (%)	11.00	22.00
Liquid limit (%)	\	18.08
Plastic limit (%)	\	32.47

**Table 2 ijerph-19-16805-t002:** Physical properties of nano-SiO_2_ [24,30].

Property	Result
Type	Silicon oxide (SiO_2_)
Purity (%)	99
APS (nm)	10–50
Bulk density (g/cm^3^)	0.10
Specific gravity	2.40
Color	Bright white

## Data Availability

Not applicable.

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
