# Peer review of "Evaluating the Effect of Nano-SiO2 on Different Types of Soils: A Multi-Scale Study"

_ijerph, 2022, doi:10.3390/ijerph192416805_

Round 1

Reviewer 1 Report

1. The structure of the introduction is not appropriate. The authors did not mention the results of many studies on the use of nanosilica in soil stabilization. What is the innovation of this research compared to previous studies? Therefore, in the introduction, the authors should mention the results of previous studies on the use of nanosilica in soil stabilization and clearly state the innovation of their research compared to previous studies in a paragraph.

The authors referenced many articles in the introduction, but they should focus on articles related to the use of nanosilica in soil stabilization. For example:

·       Mechanical properties of unbound granular materials reinforced with nanosilica (2022)

·       Combined effect of nano-silica and randomly distributed fibers on the strength behavior of clay soil (2022)

·       The impacts of nano-SiO2 and silica fume on cement kiln dust treated soil as a sustainable cement-free stabilizer (2021)

·       Effect of Nanosilica on the Macro- and Microbehavior of Dispersive Clays (2021)

·       The Influence of Nanosilica on Unconfined Compressive Strength of Frost-Susceptible Soil (2019)

·       Effects of Silica Fume and Nano-silica on the Engineering Properties of Kaolinite Clay (2018)

2. In line 35, references 4 to 11 are not needed for a simple sentence. References 15 to 17 are unnecessary for line 40.

3. On what basis was the ratio of additives chosen? The reasons should be mentioned in the paper.

4. How is the nanomaterial added to the mixture? The addition method is very important in relation to nanomaterials. This should be described in the paper.

5. Uniaxial test results show that the soil strength (qu) increases with the addition of nanosilica.

·       An increase in uniaxial strength means an increase in soil cohesion. This is in contrast to Figure 8b. What explanation do the authors have for it? This should be explained in the article.

·       Some previous studies show that the amount of nanosilica improves the strength to a certain extent, and then it has a negative effect. Why did the authors not mention this?

6. Previous studies showed that sustainability decreases if more than a certain amount of nanosilica is used. Therefore, the authors should correct lines 51-53 according to this issue. It is suggested that the authors have recommendations in this field in the conclusion section.

7. What is the reason for the brittle behavior of soil in combination with nanosilica? The authors mentioned that the presence of nanosilica does not lead to the formation of new compounds. So how do they justify the brittle behavior of the mixture?

8. Analysis of the results needs improvement. There are no sufficient scientific reasons, especially in sections 3.2 and 3.3. A comparison of the results with previous studies has not been done. Some previous studies presented different results.

9. In figures 6 and 7, determine that the values of 100, 200 and 400 are vertical stresses.

10. The manuscript contains some grammatical and language errors that need to be corrected to meet the level of the journal. For example:

Line 8: “construction, it means”: construction, which means

Line 10: “is concerned by the engineers”: is a concern for engineers

Line 23: “for the selection of soil”: for selecting soil

Line 33: “pose a threat to”: threaten

Reviewer 2 Report

(1) Purpose. What is the author's purpose in using nano-silica to solidify soil? What kind of problematic soil and what kind of problems are targeted. All of then are not introduced in the summary. Or the author can write the starting point of this article in detail.

(2) What is the correlation between experiments designed by the author? Why should we test the water loss? What is the relationship between the shear strength and UCS?

(3) When testing soil samples, the author chose two types of soil: sandy and clayey soil. Why only clayey soil is, not sandy soil , in the UCS test?

(4) Why the effect of curing conditions is tested in strength tests? What are the curing conditions?

(5) Under normal circumstances, the strength of cured soil should be higher than that of uncured soil, but why did the results in the article appear? There is no detailed explanation in the text. In addition, is there only one soil sample tested? No parallel samples for comparative analysis?

(6) Fig. 13,the vertical coordinate is wrong.

(7) What can the results of pH test prove? In combination with the full text, the addition of nano silica does not produce new minerals nor change the pH of soil, so how does its role come into being? Is there only the role of pore filling? How is pore filling verified?

Round 2

Reviewer 1 Report

1. Line 232-233: “For samples under uncured conditions (fig. 8(a)), at small content of nano- SiO2 (0.5%) addiction, the friction angle of sandy soil is promoted”

·       Figure 8 is related to clay soil.

·       By replacing figures 8a and 8b, there is no change in the existing ambiguity (comment 4). Uniaxial test results show that the clayey soil strength (qu) increases with the addition of nanosilica (Figure 10). An increase in uniaxial strength means an increase in soil cohesion. So why is there a decrease in cohesion in Figure 8 (clayey soil)?

2. Comment 6 has not been taken into account. Nanosilica in some amounts used cannot be a suitable material in terms of sustainability. Previous studies showed that sustainability decreases if more than a certain amount of nanosilica is used. Therefore, the authors should correct lines 52-54 according to this issue.

3. Why are the results of XRD not used in the analysis of the results of SEM? Only pores are considered. What is the role of chemical compounds? SEM-EDS analysis is required to detect the items shown in Figures 14 and 15. A number of researchers have reported the formation of calcium–silicate–hydrate gel. The obtained microstructural results should be compared with the results of other researchers.

·       Effect of Nanosilica on the Macro- And Microbehavior of Dispersive Clays (2021)

·       Effects of Silica Fume and Nano-silica on the Engineering Properties of Kaolinite Clay (2018)

·       The influence of nanomaterials on collapsible soil treatment (2016)

How were the samples prepared for SEM analysis? Mention the size of the magnification of the images in the text.

4. The results of the microstructural analysis should be added to the conclusion section after applying the corrections of the previous comment.

Reviewer 2 Report

The author has made considerable changes to the article, supplemented the literature and revised the charts. From the supplementary content, the overall structure is relatively complete and the argument is clear. The language has been checked and modified. The reply to the modification comments is objective.

Author Response

The language of this article has been revised according to the reviewer’s suggestion.